# Kaempferol Attenuates LPS-Induced Striatum Injury in Mice Involving Anti-Neuroinflammation, Maintaining BBB Integrity, and Down-Regulating the HMGB1/TLR4 Pathway

**DOI:** 10.3390/ijms20030491

**Published:** 2019-01-24

**Authors:** Ying-Lin Yang, Xiao Cheng, Wei-Han Li, Man Liu, Yue-Hua Wang, Guan-Hua Du

**Affiliations:** 1State Key Laboratory of Bioactive Substance and Function of Natural Medicines, Institute of Materia Medica, Chinese Academy of Medical Sciences & Peking Union Medical College, Beijing 100050, China; yinglin@imm.ac.cn; 2Beijing Key Laboratory of Drug Target Identification and New Drug Screening, Institute of Materia Medica, Chinese Academy of Medical Sciences & Peking Union Medical College, Beijing 100050, China; chengxiao@imm.ac.cn (X.C.); liwenhan@imm.ac.cn (W.-H.L.); liuman2018@yeah.net (M.L.)

**Keywords:** lipopolysaccharide (LPS), striatum, neuroinflammation, kaempferol, high mobility group box 1 (HMGB1)

## Abstract

Neuroinflammation has been demonstrated to be linked with Parkinson’s disease (PD), Alzheimer’s disease, and cerebral ischemia. Our previous investigation had identified that kaempferol (KAE) exerted protective effects on cortex neuron injured by LPS. In this study, the effects and possible mechanism of KAE on striatal dopaminergic neurons induced by LPS in mice were further investigated. The results showed that KAE improved striatal neuron injury, and increased the levels of tyrosine hydroxylase (TH) and postsynaptic density protein 95 (PSD95) in the striatum of mice. In addition, KAE inhibited the production of pro-inflammatory cytokines, including interleukin 1β (IL-1β), interleukin 6 (IL-6), tumor necrosis factor α (TNF-α), reduced the level of monocyte chemotactic protein-1 (MCP-1), intercellular cell adhesion molecule-1 (ICAM-1), and cyclooxygenase-2 (COX-2) in the striatum tissues. Furthermore, KAE protected blood-brain barrier (BBB) integrity and suppressed the activation of the HMGB1/TLR4 inflammatory pathway induced by LPS in striatum tissues of mice. In conclusion, these results suggest that KAE may have neuroprotective effects against striatum injury that is induced by LPS and the possible mechanisms are involved in anti-neuroinflammation, maintaining BBB integrity, and down-regulating the HMGB1/TLR4 pathway.

## 1. Introduction

Parkinson’s disease (PD), a devastating chronic neurodegenerative disease, is pathologically characterized by a loss of dopaminergic neurons in substantia nigra stiatum [1,2]. Tyrosine hydroxylase (TH) is involved in the synthesis of dopamine. Reduction in TH in the nigrostriatal dopamine (DA) region is a characteristic change in PD. Post-synaptic density protein 95 (PSD95) is an important scaffold protein in the synapse, which is required for synaptic plasticity.

Inflammation plays an important role in the pathogenesis of PD. It has been reported that there are astrocytes and microglial activation and lymphocytic infiltration can be seen in the brain in PD rats [3]. Others have demonstrated that anti-inflammatory drugs, such as non-steroidal anti-inflammatory drugs (NSAIDs), may have neuroprotective effects in PD patients and animals [4,5]. Microglia activating and losing dopaminergic neurons induced by lipopolysaccharide (LPS) can be seen in the striatum [6]. High-mobility group box 1 (HMGB1) can be passively released by necrotic cells and actively secreted by inflammatory cells [7]. In the normal condition, HMGB1 exists as a nuclear protein, while under pathological condition displays an inflammatory cytokine-like activity in the extracellular space [8]. Some actions of HMGB1 are mediated through the toll-like receptors (TLRs) [9]. It is reported that interaction between HMGB1 and TLR4 leads to increased production and the release of cytokines via upregulation of NF-κB [10,11].

The blood-brain barrier (BBB) is formed by high resistance tight junctions (TJ) within the capillary endothelium. Normal BBB maintains a homeostatic microenvironment of the central nervous system (CNS). Under pathological conditions, such as neuroinflammation, monocyte chemoattractant protein-1 (MCP-1), and intercellular adhesion molecule 1 (ICAM-1) that are transiently and significantly up-regulated, are able to disrupt the integrity of BBB [12,13]. Overactivated microglia cells can produce pro-inflammatory factors, nitric oxide, and reactive oxygen species (ROS), which can damage the BBB and induce neuron damage [14]. In endothelial cells, recombinant human HMGB1 elicited pro-inflammatory responses, which subsequently resulted in an increase of vascular permeability, cell swelling, and disruption of BBB [15].

Kaempferol (KAE), derived from the roots of the ginger plant Kaempferol galanga L., has a variety of biological activities, such as anti-oxidant, anti-inflammatory, anti-cancer, and anti-angiogenesis [16]. Studies have already demonstrated kaempferol can exert protective effects and anti-inflammation after stimulating by LPS in BV2 cells [16]. Furthermore, our previous study had demonstrated that kaempferol alleviates neuroinflammation in the cortex of mice injured by LPS [17]. In the present study, the effects and possible mechanism of kaempferol on LPS-induced striatum injury in mice were investigated.

## 2. Results

### 2.1. Kaempferol Protects Striatum Neuron Injuried by LPS in Mice

The ultrastructures of the striatum tissues were observed by electron microscopy. In the control group, the nucleolus exhibited a clear boundary, and the nuclei of neuronal cells appeared large and round (Figure 1Aa). Conversely, after LPS injury the neurons were markedly swollen and exhibited signs of apoptosis, such as nuclear envelope shrinkage (Figure 1Ab). However, the neuron damage improved in the KAE treatment group (Figure 1Ac,d). In addition, when compared with the control group, LPS reduced the expression of TH and PSD95 in the striatum (both *p* < 0.01). However, KAE can significantly improve the expression level of TH (Figure 1B) and PSD95 (Figure 1C) as compared with the LPS-induced group (*p* < 0.01).

### 2.2. Kaempferol Inhibits Microglia Activation in Striatum of Mice Injured by LPS

To observe whether KAE inhibits the activation of microglia, we performed immunofluorescence staining by Iba-1 for KAE 50 mg/kg group. Treatment of mice with LPS increased the levels of Iba-1 both in ventral striatum (Figure 2A) and in dorsal striatum (Figure 2B). KAE treatment inhibited the expression of Iba-1 in the striatum tissues of LPS-injured mice. This finding indicates that KAE inhibits the activation of Iba-1 following stimulation of mice by LPS.

### 2.3. Kaempferol Blocked BBB Dysfunction Injured by LPS in the Striatum of Mice

The blood-brain barrier is composed of a microvascular endothelium and intercellular tight junction, astrocytes, and basement membrane. BBB integrity by Evance Blue staining was investigated in our previous article [17]. Therefore, in this study, we focused on the ultrastructure of BBB observed using electron microscopy and tight junction proteins determined by WB. In the control group, the endothelial cells were closely interconnected and no edema was detected in the area surrounding the capillaries (Figure 3Aa). Conversely, the endothelial cells protruded towards the cavity and the capillaries were shrunken and deformed in the LPS-injured group. The extension of the astrocytes in the BBB swelled and vesicles were formed (Figure 3Ab). However, KAE treatment markedly attenuated ultrastructure destruction of BBB injured by LPS in the striatum tissue and improved the edema of the astrocytes and the area surrounding the capillaries (Figure 3Ac and Figure 3Ad). In addition, as the tight junction proteins, such as occludin, claudin-1, and CX-43, play important roles in maintain BBB integrity; these proteins were also examined in the striatum. In LPS-injured mice, levels of occludin (Figure 3B, *p* < 0.01), claudin-1 (Figure 3C, *p* < 0.01), and CX-43 (Figure 3D, *p* < 0.01) in the striatum of mice were all decreased significantly when compared with the control group. However, treatment with KAE 20 mg/kg and 50 mg/kg both significantly improved the proteins expression of claudin-1, occludin, and CX-43. These results suggest that KAE can reduce BBB damage injured by LPS to a certain extent.

### 2.4. Kaempferol Inhibits Inflammatory Cytokines Release in Striatum of Mice Injured by LPS

Recognizing that BBB compromise is associated with inflammation in the striatum, we further analyzed expression of inflammatory cytokines, chemokine, cell adhesion molecule, and cyclooxygenase-2 (COX-2). LPS stimulated the release of interleukin 1β (IL-1β) (Figure 4A, *p* < 0.01), interleukin 6 (IL-6) (Figure 4B, *p* < 0.01), tumor necrosis factor α (TNFα) (Figure 4C, *p* < 0.01), monocyte chemotactic protein-1 (MCP-1) (Figure 4D, *p* < 0.01), ICAM-1 (Figure 4E, *p* < 0.01), and COX-2 (Figure 4F, *p* < 0.01). However, KAE dose-dependently suppressed the production of pro-inflammatory cytokines and chemokines involving IL-1β, IL-6, TNFα, MCP-1, ICAM-1, and COX-2 injured by LPS in the striatum of mice.

### 2.5. Kaempferol Blocked the Expression of Inflammatory Proteins in Striatum of Mice Injured by LPS

LPS injury significantly increased the level of HMGB1and TLR4 both on the mRNA level and the protein level (all *p* < 0.01, Figure 5) in the striatum of mice. However, KAE administration inhibited the expression of HMGB1 and TLR4 proteins. These finding indicates that KAE down-regulated the HMGB1/TLR4 inflammatory pathway in striatum of mice.

## 3. Discussion

Kaempferol, a flavonoid, plays a variety of pharmacological activities [17]. Our previous investigation identified that kaempferol exerted protective effects on cortex neurons that were injured by LPS. In this study, the effects and possible mechanism of kaempferol on the striatum neuron injured by LPS in mice were investigated. The results showed kaempferol protects against LPS-induced striatum injury, and the possible mechanisms involve anti-neuroinflammation, maintaining BBB integrity, and down-regulating HMGB1/TLR4 inflammatory pathway.

Parkinson’s disease (PD) is an aging-related movement disorder in the striatum of the brain [18]. Tyrosine hydroxylase (TH) catalyzes the conversion of l-tyrosine to l-3,4-dihydroxyphenylalanine (L-DOPA), which is the initial and rate-limiting step in the biosynthesis of DA. TH was a synthetic rate limiting enzyme of dopaminergic neurons, which was positively correlated with dopaminergic neurons. Accordingly, TH has been speculated to play some important roles in the pathophysiology of PD [19]. Thus, promoting the proliferation of TH-positive cells would help in the prevention and treatment of PD. PSD95, which is highly expressed in the cerebral cortex, hippocampus, and striatum, is an important molecule on the postsynaptic membrane that plays a pivotal role in learning and memory formation [20]. In the present study, mice that were treated with kaempferol significantly improved the levels of both TH and PSD95 decreased by LPS in the striatum of mice. This suggests that kaempferol treatment markedly improved striatum injury and promoted synaptogenesis and synaptic plasticity.

In the CNS, BBB prevents nonspecific export of large and/or polar molecules. The BBB of the striatum is especially fragile, some stimuli such as ischemic, osmotic, or other stressors increased BBB permeability appears in the striatum before any other brain region [21]. In recent reports, BBB dysfunction in the striatum of PD patients was reported [22]. In 1-Methyl-4-phenyl-1,2,3,6-tetrahydropyridin (MPTP) and 6-hydroxydopamine (6-OHDA) PD mouse models also show impaired striatal BBB integrity with the disruption of tight junctions [23,24]. Occludin is an integral plasma-membrane protein that is located at the tight junctions, described for the first time in 1993 by Shoichiro Tsukita [25]. Studies have shown that occludin is an important protein in tight junction stability and barrier function [26]. Claudins, along with occludin, are the most important components of the TJ, which act as major constituents of the tight junction complexes that regulate the permeability of epithelia. Connexin 43 (CX-43) is a protein that in humans is encoded by the Gap junction alpha-1 protein (GJA1) gene on chromosome 6 [27]. In the present study, our results showed that LPS injury resulted in BBB disruption and the proteins of claudin-1, occluding, and CX-43 were all decreased significantly in comparison with normal mice. However, kaempferol significantly improved the levels of claudin-1, occluding, and CX-43 decreased by LPS in the striatum of mice. This suggests that kaempferol treatment markedly improved BBB disruption in the striatum. Our results that were observed by electron microscopy also approved that kaempferol treatment significantly improved BBB disruption injured by LPS, such as endothelial cells protruding into the lumen, capillary contraction and deformation, and astrocyte expansion.

In recent years, neuroinflammation has emerged as a critical contributor to PD pathogenesis. Reactive microglia and inflammatory molecules have been found in the striatum of PD patients, and several cytokine polymorphisms have been reported as risk factors for PD onset [28]. Upon CNS injuries, activated microglia cells can change into an amoeboid morphology and release inflammatory factors. Microglial suppression is considered to be a key strategy in the search for neuroprotection [29,30]. Studies on the brains of the postmortem PD patients have shown the presence of activated microglia cells in the substantia nigra pars compacta [31] and revealed an elevation of inflammatory regulators, including inducible nitric oxide synthase (iNOS) and cyclooxygenase-2 (COX-2) expression in the striatum area [32]. In an animal model of PD, employing intrastriatal injection of 6-hydroxydopmaine (6-OHDA), activated microglia were identified in striatum [33]. The blood–brain barrier (BBB) is a dynamic structure that maintains the homeostasis of the brain and thus proper neurological functions. Perivascular cells, including microglial cells, astrocytes, and macrophages and brain microvascular endothelial cells (BMEC) produce various inflammatory factors that affect the BBB permeability and the expression of adhesion molecules [13]. The chemokine CCL2/MCP-1 binds to CCR2 on brain ECs to initiate intracellular signaling cascades that result in the dynamic reorganization of junction complexes and EC retraction. CCL2/MCP-1 induces the phosphorylation of TJ proteins, resulting in their disassembly and/or redistribution from the cell border. During inflammation, monocyte chemoattractant protein-1 (MCP1) is up-regulated to disrupt the integrity of BBB and modulate the progression of various diseases [11]. ICAM-1, an endothelial- and leukocyte-associated transmembrane protein, plays an important role in stabilizing cell-cell interactions and facilitating leukocyte endothelial transmigration [12]. In BMEC, cytokines can stimulate the expression of several adhesion molecules. Among these adhesion molecules, the intercellular adhesion molecule-1 (ICAM-1) binds to its leukocyte ligands and allows for activated leukocytes entry into the CNS. In the present study, to evaluate the effect of kaempferol on LPS-induced microglia activation in vivo, the expression of Ibal-1, the microglia marker, was observed by immunohistochemistry. The results showed that the morphology of microglia was changed from a ramified to an amoeboid shape induced by LPS, while kaempferol treatment reversed it. In addition, kaempferol inhibited the production of pro-inflammation cytokines, including IL-1β, IL-6, TNF-α, MCP-1, and ICAM-1 in striatum tissues. These results suggest that kaempferol treatment significantly alleviates microglia activation and inflammatory responses in the striatum of mice that are injured by LPS.

HMGB1, a highly conserved nonhistone nuclear protein, can be passively released from damaged cell to exacerbate inflammation [7]. In the nucleus, HMGB1 regulates several interactions of transcription factors with DNA as well as contributing to genome stability and DNA repair [34]. Several recent studies have supported that HMGB1 exhibits a critical role in PD-associated neuroinflammation. In the CSF and serum of PD patients, elevated HMGB1 protein levels have been detected in the post-mortem midbrain tissues [35]. Therefore, pharmacological targeting of HMGB1 could be a reasonable approach to attack PD progression. It has reported that the anti-inflammatory and anti-degenerative effects of anti-HMGB1 mAbs should be a rational treatment approach in a rat model of PD [8] and in MPTP-induced neurotoxicity in mice [35]. These results demonstrate that HMGB1 serves as a powerful bridge between progressive dopaminergic neurodegeneration and chronic neuroinflammation in PD, suggesting that HMGB1 is a suitable target for neuroprotective trials in PD. In neuron-glial cultures, HMGB1 has been shown to activate microglia to release inflammatory molecules, such as IL-1β and IL-6. Moreover, in the anti-HMGB1-treated group, the disruption of BBB and the elevated vascular permeability caused by 6-OHDA neurotoxicity was inhibited. Therefore, along with the direct suppression of cytokines in the rat brain, anti-HMGB1 administration prevented leakage in BBB, which is proposed to indirectly contribute to PD pathogenesis [36]. In addition, HMGB1 has been shown to participate in modulate autophagy and apoptosis as well as regulate gene transcription [37,38]. Extracellular HMGB1 can then interact with various receptors including the receptor for advanced glycation end products (RAGE) and TLR2, TLR4, and TLR9. TLR4, one of the main receptors of HMGB1, play an important role in HMGB1-induced neuroinflammation [39]. Our results also suggest that kaempferol treatment markedly down-regulated LPS-induced inflammatory proteins expression, including HMGB1 and TLR4.

In summary, kaempferol exerts neuroprotection on the striatum of mice injured by LPS and the possible mechanisms are involved in anti-neuroinflammation, maintaining BBB integrity, and down-regulating the HMGB1/TLR4 pathway. Though the present study found that kaempferol treatment downregulated the HMGB1 protein and TLR4 expression, there is a lack of in-depth study, such as the translocation of HMGB1 after LPS induction and after kaempferol treatment. In the latter research, the translocation of HMGB1 after LPS induction and kaempferol treatment will be further explored. Moreover, the distribution and regulation mechanism of HMGB1 in different regions of brain, plasma, serum, or CSF will be further investigated.

## 4. Material and Methods

### 4.1. Reagents and Antibody

Lipopolysaccharide was purchased from Sigma-Aldrich Co. (St. Louis, MO, USA). IL-1β, IL-6, TNFα, MCP-1, and ICAM-1 ELISA kits were purchased from Genetimes Technology Inc. (Shanghai, China). Anti-TH antibody, anti-PSD-95 antibody, anti-COX-2 antibody, anti-Claudin-1 antibody, anti-Occludin antibody, and anti-CX43 antibody, anti-HMGB1 antibody, anti-TLR4 antibody, and Cy3-conjugated secondary antibody were the products of Abcam (Cambridge, United Kingdom). Anti-Iba-1 was the product of Wako (Osaka, Japan).

### 4.2. Animals and Experimental Procedure

Adult male BALB/c mice weighing 18 to 22 g purchased from Laboratory Animal Centre of Beijing Hua-Fu-Kang Bioscience Co., LTD (Beijing, China; SCXK (Jing) 2014-0004) were randomly divided into following groups (12 mice each group): Control group, LPS model group (pre-treatment with vehicle for 7 d and LPS 5 mg/kg by i.p. on the 7th day), LPS+KAE 20 group (pre-treatment with kaempferol 20 mg/kg for 7 d and LPS 5mg/kg by i.p. on the 7th day), and LPS+KAE 50 group (pre-treatment with kaempferol 50 mg/kg for 7 d and LPS 5mg/kg by i.p. on the 7th day) [17]. Furthermore, all animal care and experimental procedures regarding the animals were approved by the ethic committees of Institute of Materia Medica, Chinese Academy of Medical Sciences & Peking Union Medical College (No.11401300068570, Oct. 10, 2017).

### 4.3. Immunofluorescence Staining

After treatment, the mice were perfused with 0.1 M phosphate-buffered saline and followed by 4% paraformaldehyde for immunofluorescence staining [17]. Briefly, sections were incubated with anti-Iba-1 primary antibody (1:200) for overnight at 4 °C in blocking solution. After washing with PBS, the brain sections were incubated with Cy3-conjugated secondary antibody for 2 h at room temperature. Nuclei were stained with 4,6-diamidino-2-phenylindole (DAPI) for 10 min at room temperature. The fluorescence microscopy was used to observe the staining results.

### 4.4. Electron Microscopy Analysis

The ultrastructures of the striatum tissues were investigated using electron microscopy [40,41]. Briefly, the striatum tissues of mice were collected and fixed initially with 2.5% glutaraldehyde and then post fixed in 1% osmium tetroxide. After fixing, the tissues were dehydrated with gradient mixtures of ethanol and acetone and embedded in Epon 812. Subsequently, the tissues were cut into 1 mm^3^ sections and double-stained with uranyl acetate and lead nitrate. Tissue samples were observed under a transmission electron microscope.

### 4.5. ELISA Assay

Six hours after LPS application, the striatum tissues of mice (8 mice each group) were obtained and prepared by homogenizer in cool saline. Subsequently, the supernatant, after centrifuging at 4500 rpm for 10 min at 4 °C, was used to detect the level of IL-1β, IL-6, TNF-α, MCP-1, and ICAM-1 by ELISA kits, according to the introductions of the manufacturer.

### 4.6. Western Blot Analysis

Six hours after LPS application, the striatum tissues of mice were removed and stored at −80 °C for further detection. Frozen striatum tissues were homogenized in cool RIPA buffer with the cocktail protease inhibitor. After that, they were lysed on the ice for 30 min. Afterwards, the supernatants were collected and quantified by BCA assay after centrifuging at 12,000 rpm for 10 min at 4 °C. Proteins were separated by 10% SDS-PAGE gel and transferred on the PVDF membranes. The membranes were incubated in 5% BSA solution in order to occupy the nonspecific sites on the membranes at 37 °C. Next, the membranes were incubated with anti-TH antibody, anti-PSD-95 antibody, anti-COX-2 antibody, anti-Claudin-1 antibody, anit-Occuldin-1 antibody, anti-CX43 antibody, anti-HMGB1 antibody, and anti-TLR4 antibody respectively at 4 °C overnight. After horseradish peroxidase-conjugated secondary antibody incubation. The signal densities on the blots were measured using the enhanced ECL system and normalized using an internal control. The results from animals under various experiment conditions then were normalized by value of the corresponding control animal (fold change relative to control).

### 4.7. Real-Time PCR Analysis

TRIzol method was used to extract RNA, as described previously [17]. The Primers included HMGB1 (forward, GGCTGACAAGGCTCGTTATG; reverse, GGGCGGTACTCAGAACAGAA), TLR4 (forward, CTCTGGGGAGGCACATCTTC; reverse, CAGGTCCAAGTTGCCGTTTC), and β-actin (forward, AGGCCAACCGTGAAAAGATG; reverse, TGGCGTGAGGGAGAGCATAG). Relative quantitation was determined using real-time SYBR green fluorescence and then calculated by means of the comparative Ct method (2^−ΔΔ*C*t^) with the expression of β-actin as an internal control.

### 4.8. Statistical Analysis

Statistical analysis was performed using GraphPad Prism 6.02 (GraphPad Software Inc., CA, USA) Data are expressed as the mean ± SD. Measurement data between groups were compared using one-way analysis of variance and the Student-Newman-Keuls test. *p* < 0.05 was considered to be statistically significant.

## Figures and Tables

**Figure 1 ijms-20-00491-f001:**
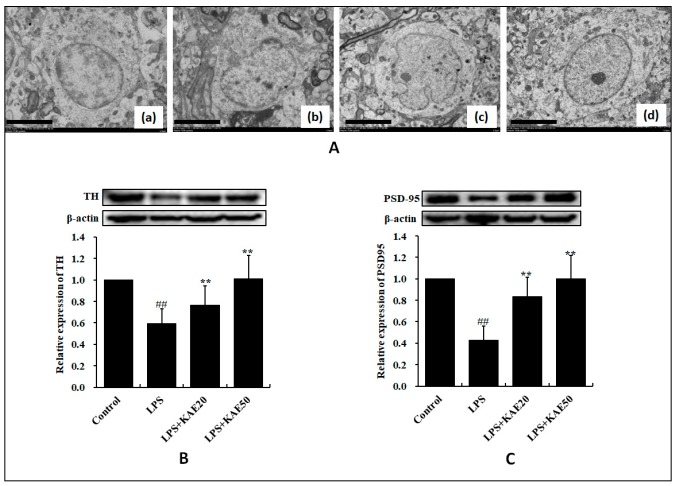
Effect of kaempferol on the striatum of of mice injured by lipopolysaccharide (LPS). (**A**) Electron microscopy analysis of the ultrastructure alterations in striatum derived from the various treatment groups: control group (**a**), LPS group (**b**), LPS+KAE 20 mg/kg group (**c**); LPS+KAE 50 mg/kg group (**d**); (**B**) Expression of TH protein; and (**C**) Expression of PSD-95 protein. Values are mean ± SD (*n* = 4). ^##^
*p* < 0.01 vs. Control group. ** *p* < 0.01 vs. LPS group. Scale bars 5 μm in (**a**–**d**), *n* = 3 per group.

**Figure 2 ijms-20-00491-f002:**
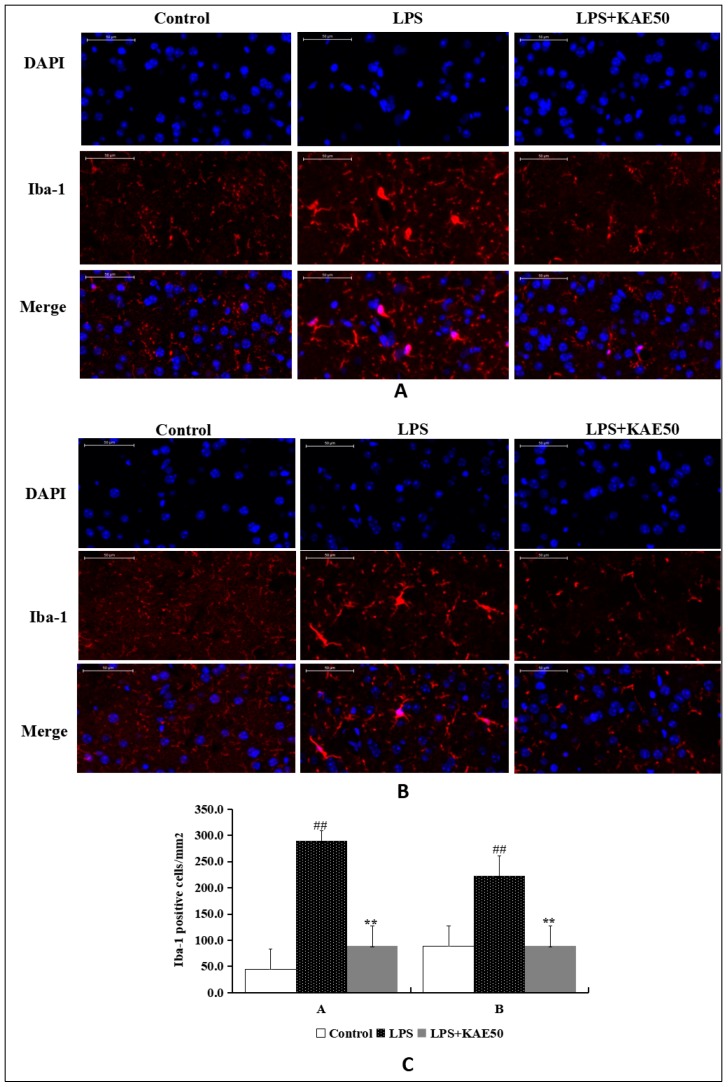
Effect of kaempferol on microglial activation stained by immunohistochemistry with anti-Iba-1 antibody. (**A**) Representative image in ventral striatum of mice; (**B**) Representative image in dorsal striatum of mice; and, (**C**) The quantitative analysis of Iba-1 positive cells. Values are mean ± SD (*n* = 3). ^##^
*p* < 0.01 vs. control group. ** *p* < 0.01 vs. LPS group. Scale bars 50 μm in (**A**,**B**).

**Figure 3 ijms-20-00491-f003:**
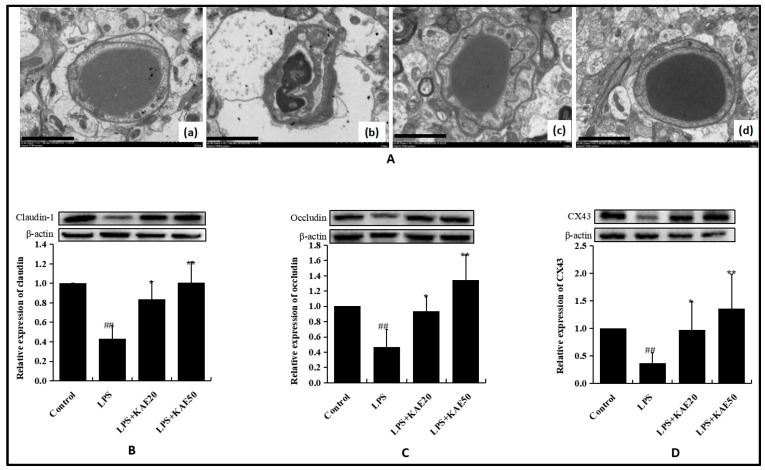
Effect of kaempferol on the ultrastructure and tight junction proteins of blood–brain barrier (BBB) in the striatum of mice injured by LPS. (**A**) BBB ultrastructure observed by electronic microscope: control group (**a**), LPS group (**b**), LPS+KAE 20 mg/kg group (**c**); LPS+KAE 50 mg/kg group (**d**); (**B**) the expression level of Claudin-1; (**C**) the expression level of Occludin; and, (**D**) the expression level of CX43. Values are mean ± SD (*n* = 4). ^##^
*p* < 0.01 vs. control group. * *p* < 0.05, ** *p* < 0.01 vs. LPS group. Scale bars 5 μm in (**a**–**d**), *n* = 3 per group.

**Figure 4 ijms-20-00491-f004:**
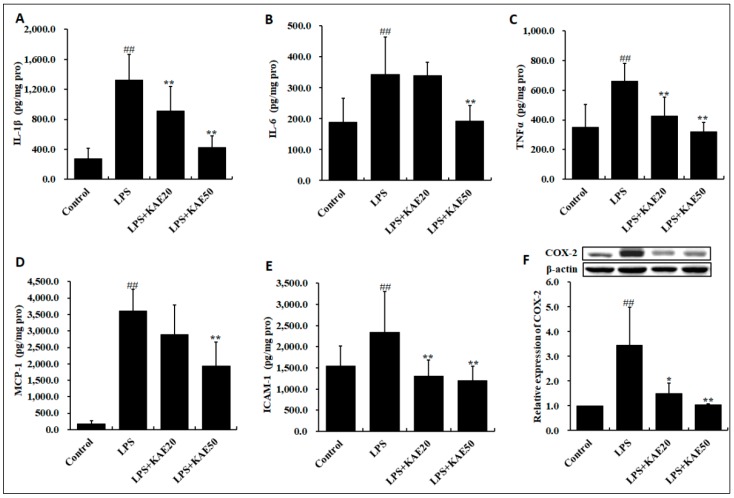
Effect of kaempferol on pro-inflammatory factors and chemokines in the striatum of mice injured by LPS. (**A**) interleukin 1β (IL-1β); (**B**) interleukin 6 (IL-6); (**C**) tumor necrosis factor α (TNF-α); (**D**) monocyte chemotactic protein-1 (MCP-1); (**E**) intercellular adhesion molecule-1 (ICAM-1); and, (**F**) cyclooxygenase-2 (COX-2). Values are expressed as mean ± SD (*n* = 8). ^##^
*p* < 0.01 vs. Control group. * *p* < 0.05, ** *p* < 0.01 vs. LPS group.

**Figure 5 ijms-20-00491-f005:**
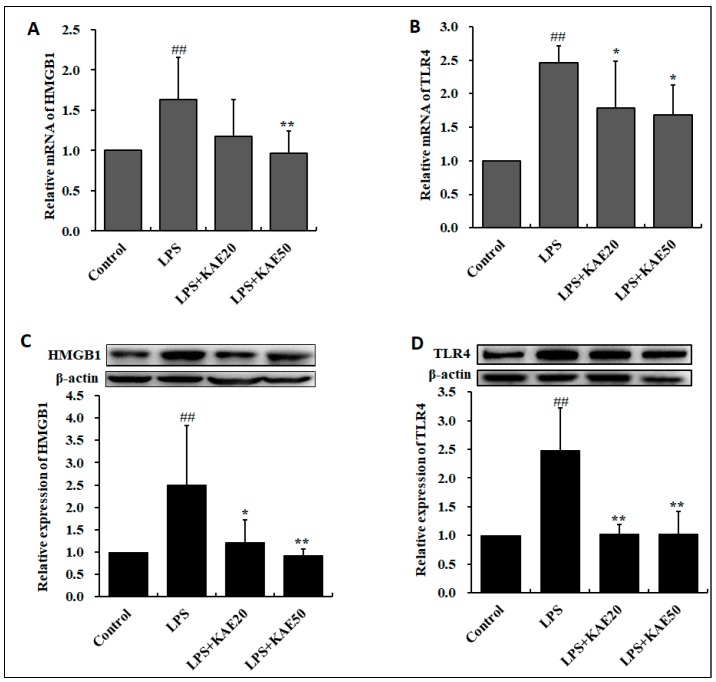
Effect of kaempferol on mRNA and protein expression of HMGB1 and TLR4 in the striatum of mice injured by LPS. (**A**) mRNA expression of HMGB1; (**B**) mRNA expression of TLR4; (**C**) protein expression of HMGB1; and, (**D**) protein expression of TLR4. Values are mean ± SD (*n* = 4). ^##^
*p* < 0.01 vs. Control group; * *p* < 0.05, ** *p* < 0.01 vs. LPS group.

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
