# Peer review of "Kaempferol Attenuates LPS-Induced Striatum Injury in Mice Involving Anti-Neuroinflammation, Maintaining BBB Integrity, and Down-Regulating the HMGB1/TLR4 Pathway"

_ijms, 2019, doi:10.3390/ijms20030491_

Round 1
Reviewer 1 Report
Authors have precisely elucidated the different facets of Kaempferol (KAE) in an experimental model of LPS induced striatum injury. However, I have some concerns and suggestions regarding the manuscript.
1) In the introduction section authors have related more on Parkinson's disease (PD), hence I would suggest authors to follow this recent paper and include an overview regarding the role of HMGB1 in PD.
Angelopoulou, Efthalia, Christina Piperi, and Athanasios G. Papavassiliou. "High‐mobility Group Box 1 in Parkinson's disease: from pathogenesis to therapeutic approaches." Journal of neurochemistry (2018).
2) My major concerns is regarding the earlier reported finding on Kaempferol by the same group.
In that study authors have already demonstrated the various role of Kaempferol in the similar model (LPS). Authors have already stated that Kaempferol exhibited anti-neuroinflammatory effects, protects BBB permeability and inhibits HMGB1 release by downregulating TLR4/MyD88 pathway.
(Cheng, X., Yang, Y. L., Yang, H., Wang, Y. H., & Du, G. H. (2018). Kaempferol alleviates LPS-induced neuroinflammation and BBB dysfunction in mice via inhibiting HMGB1 release and down-regulating TLR4/MyD88 pathway. International immunopharmacology, 56, 29-35.)
I am not seeing any major difference between two studies except experimental design where in earlier studies author have investigated the dose of Kaempferol up to 100 mg/kg which is up to 50 mg /kg in current study. Current study focused against striatum whereas earlier focused on cortex and hippocampus
Similar findings on Kaempferol by same group in the current manuscript raises my concerns about the novelty of the study. How will authors defends about the novelty of current study?
3) I did not find any data regarding the dose of LPS throughout the manuscript. Though author have mentioned it was done as per their earlier reported paper, but I will recommend author to state the dose of LPS.
4) Authors state that Kaempferol treatment downregulated the HMGB1 protein and TLR4 expression. Did authors have investigated the translocation of HMGB1 after LPS induction and after Kaempferol treatment?
5) Did authors separately investigated the expression level of HMGB1 and TLR4 except in the striatum (plasma, serum or CSF)?
As findings are emerging that, during any neuronal injury due to translocation of HMGB1 from nuclei to surrounding area or extracellular fluid there is decreased brain level of HMGB1 and increased serum or plasma level of HMGB1.
(Fu, Li, Keyue Liu, Hidenori Wake, Kiyoshi Teshigawara, Tadashi Yoshino, Hideo Takahashi, Shuji Mori, and Masahiro Nishibori. "Therapeutic effects of anti-HMGB1 monoclonal antibody on pilocarpine-induced status epilepticus in mice." Scientific Reports 7, no. 1 (2017): 1179.)
However, there is increased brain level of HMGB1 after LPS administration. Please justify?
Author Response
Response to Reviewer 1 Comments
Point 1: In the introduction section authors have related more on Parkinson's disease (PD), hence I would suggest authors to follow this recent paper and include an overview regarding the role of HMGB1 in PD. Angelopoulou, Efthalia, Christina Piperi, and Athanasios G. Papavassiliou. "High‐mobility Group Box 1 in Parkinson's disease: from pathogenesis to therapeutic approaches." Journal of neurochemistry (2018).
Response 1: Thank you for your suggestion. This article is quoted in the discussion section (See Ref. 38).
Point 2: My major concern is regarding the earlier reported finding on Kaempferol by the same group. In that study authors have already demonstrated the various role of Kaempferol in the similar model (LPS). Authors have already stated that Kaempferol exhibited anti-neuroinflammatory effects, protects BBB permeability and inhibits HMGB1 release by downregulating TLR4/MyD88 pathway. (Cheng, X., Yang, Y. L., Yang, H., Wang, Y. H., & Du, G. H. (2018). Kaempferol alleviates LPS-induced neuroinflammation and BBB dysfunction in mice via inhibiting HMGB1 release and down-regulating TLR4/MyD88 pathway. International immunopharmacology, 56, 29-35.). I am not seeing any major difference between two studies except experimental design where in earlier studies author have investigated the dose of Kaempferol up to 100 mg/kg which is up to 50 mg /kg in current study. Current study focused against striatum whereas earlier focused on cortex and hippocampus. Similar findings on Kaempferol by same group in the current manuscript raises my concerns about the novelty of the study. How will authors defend about the novelty of current study?
Response 2: Our previous article (Int. Immunopharmacol., 2018, 56, 29-35.) demonstrated that Kaempferol exerted protective effects on cortex and hippocampal neuron injured by LPS. However, in the present study, the effects of Kaempferol on striatal dopaminergic neurons induced by LPS in mice were investigated.
Point 3: I did not find any data regarding the dose of LPS throughout the manuscript. Though author have mentioned it was done as per their earlier reported paper, but I will recommend author to state the dose of LPS.
Response 3: The dose of LPS was added in Methods section in the revised manuscript.
Point 4: Authors state that Kaempferol treatment downregulated the HMGB1 protein and TLR4 expression. Did authors have investigated the translocation of HMGB1 after LPS induction and after Kaempferol treatment?
Response 4: In the latter study, the translocation of HMGB1 after LPS induction and Kaempferol treatment will be further explored.
Point 5: Did authors separately investigated the expression level of HMGB1 and TLR4 except in the striatum (plasma, serum or CSF)? As findings are emerging that, during any neuronal injury due to translocation of HMGB1 from nuclei to surrounding area or extracellular fluid there is decreased brain level of HMGB1 and increased serum or plasma level of HMGB1. (Fu, Li, Keyue Liu, Hidenori Wake, Kiyoshi Teshigawara, Tadashi Yoshino, Hideo Takahashi, Shuji Mori, and Masahiro Nishibori. "Therapeutic effects of anti-HMGB1 monoclonal antibody on pilocarpine-induced status epilepticus in mice." Scientific Reports 7, no. 1 (2017): 1179.). However, there is increased brain level of HMGB1 after LPS administration. Please justify?
Response 5: In the latter study, the distribution and regulation mechanism of HMGB1 in different region of brain, plasma, serum or CSF will be further explored.
Reviewer 2 Report
In general study shows attenuation of LPS-induced toxicity by kaempfenol. The study demonstrate the role of this drug in neuroinflammation and BBB integrity. I have several comments to improve the study.
Introduction, first paragraph: This article has little relevance to PD. Introduction should be focused on LPS-induced neuroinflammation. Relevance to PD should be moved to Discussion.
Fig.2. Quantification for Iba1 staining should be provided
Fig.3. BBB integrity should be also investigated by traditional methods such as Evance Blue staining.
Author Response
Point 1: Introduction, first paragraph: This article has little relevance to PD. Introduction should be focused on LPS-induced neuroinflammation. Relevance to PD should be moved to Discussion.
Response 1: Thank you for your suggestion. The introduction section was revised in the revised manuscript.
Point 2: Fig.2. Quantification for Iba1 staining should be provided
Response 2: In this study, microglia activation was observed qualitatively after KAE treatment only at a dose of 50mg/kg.
Point 3: Fig.3. BBB integrity should be also investigated by traditional methods such as Evance Blue staining.
Response 3: BBB integrity by Evance Blue staining was investigated in our previous article. (Int. Immunopharmacol. 2018, 56: 29-35.). Therefore, in this study, we focused on the BBB integrity in the striatal tissues. The ultrastructure of BBB was observed using electron microscopy and tight junction proteins was determined by WB.
Reviewer 3 Report
The paper entitled “Kaempferol attenuates LPS-induced striatum injury in mice involving anti-neuroinflammation, maintaining BBB integrity and down-regulating HMGB1/TLR4 pathway” aims to evaluate role of kaempferol in the striatum of adult mice after LPS-induced neuroinflammation. The authors used electron microscopy, immunohistochemistry, ELISA as well as mRNA and protein experiments to confirm that kaempferol reduces LPS-induced damage in the striatum of adult mice. The manuscript, however, has a several major and minor problems.
Major criticisms:
(1) The English language shows considerable deficiencies and should therefore be checked urgently by a native speaker. For example:
a. Line 14: “… Our previous investigation had identified that kaempferol (KAE) exerted protective effects on cortex neurons injured by LPS…”
b. Line 18: “… In addition, KAE inhibited the production of pro-inflammation pro-inflammatory cytokines including…”
c. Line 50: “…Overactive Overactivated microglia cell can product produce substantial…”
d. Line 120: “…Treatment of mice with LPS significantly increased the level of inflammatory proteins including HMGB1and TLR4 both in on the mRNA level and protein level…”. Moreover, there is a missing space at “HMGB1and”.
e. Line 132: “…In this study, the effects and possible mechanism of kaempferol on the striatum induced by LPS in mice were also investigated…” Please restructure this sentence.
(2) Please be careful that MCP-1, ICAM-1 and COX-2 are no cytokines (line 19, following; line 110 following).
(3) Please restructure the introduction to create a better connection between the performed analyses and the topic. For example, why is it important to you to analyse MCP-1 and ICAM-1?
(4) Why do your control groups have no standard derivation in cause of western blot and gene expression experiments?
(5) Please add a quantitative analysis in bars for the immunohistochemistry experiments. Moreover, add the statistical analysis when claiming that “… KAE treatment inhibited the expression of Iba-1 in the striatum tissues of LPS-induced mice. This finding indicates that KAE inhibits the expression of Iba-1 following stimulation of mice by LPS…”, as you only show the results of one mouse. In addition, please add a size bar or magnification factor for this figure.
(6) Why did you not perform immunohistochemistry with mice that received KAE25?
(7) Please explain your animal model briefly instead of only referring to another publication. How many animal were used for the whole experiment? Why differ the “n” between 4 and 8? How was KAE dissolved? How were LPS and KAE applied? Please add that to your methods.
(8) What was the rationale to isolate proteins for ELISA with saline (4.5.) and for western blot with RIPA buffer (4.6.)?
(9) Please connect your discussion more detailed to the investigated parameters and analyses you performed. For example:
a. Discuss the difference between the two concentrations you used. Which one is “better” and why?
b. Does KAE have some negative side effects?
c. Why did you pre-treat the animals for seven days with KAE? Is it necessary for KAE to exert its protective effects? Are there other studies with different application time points or even just with one time application?
d. Please add limitations to your study.
Minor criticisms:
Please announce the abbreviation “DA” (line 35) an MPTP (line 151).
Line 55: “…Kaempferol (KAE) has a variety of biological activities, such as anti-oxidant, anti-inflammatory, clearing the free radicals, anti-cancer, and anti-angiogenesis, and plays an important role in cancer, inflammation and cardiovascular diseases…”.
Please delete “clearing free radicals” as this is included in “anti-oxidant”.
The same for “anti-cancer” and “plays an important role in cancer”.
Please check for the size of the font, as it differs several times.
Pease add a size bar or magnification factor for figure 1 A (a-d) and 3 A (a-d).
Figure 1: Please use “A,B,C”, like in the description below, instead of “A,A,B”. Adjust the letters also in line 70 and 71.
Figure 2: Please describe A and B once again in the description of figure 2.
Figure 3: You used “B” twice in the description of this figure. Please change that.
Line 97: Please change “occluding” to “occludin”.
Please choose a more representative western blot for the quantitative results of HMGB1 (figure 5C). The bars of the control and LPS+KAE20 look nearly the same, but the representative western blots were nearly showing the opposite.
Line 237: “…After LPS stimulating for 6 hours…”. Please change that into “Six hour after LPS application”, as otherwise, it sounds like there was a continuous LPS applicative for six hours.
Pease add the used internal control for western blot analyses in section 4.6 and be careful that –ΔΔCt is superscripted.
Author Response
Major criticisms:
Point 1: The English language shows considerable deficiencies and should therefore be checked urgently by a native speaker. For example:
a. Line 14: “… Our previous investigation had identified that kaempferol (KAE) exerted protective effects on cortex neurons injured by LPS…”
b. Line 18: “… In addition, KAE inhibited the production of pro-inflammation pro-inflammatory cytokines including…”
c. Line 50: “…Overactive Overactivated microglia cell can product produce substantial…”
d. Line 120: “…Treatment of mice with LPS significantly increased the level of inflammatory proteins including HMGB1and TLR4 both in on the mRNA level and protein level…”. Moreover, there is a missing space at “HMGB1and”.
e. Line 132: “…In this study, the effects and possible mechanism of kaempferol on the striatum induced by LPS in mice were also investigated…” Please restructure this sentence.
Response 1: These English languages were checked and revised carefully.
Point 2: Please be careful that MCP-1, ICAM-1 and COX-2 are no cytokines (line 19, following; line 110 following).
Response 2: The expression of this sentence was revised in the revised manuscript.
Point 3: Please restructure the introduction to create a better connection between the performed analyses and the topic. For example, why is it important to you to analyse MCP-1 and ICAM-1?
Response 3: In the introduction, significances of MCP-1 and ICAM-1 detection were added according to the suggestion. In addition, relevant content was added to the discussion.
Point 4: Why do your control groups have no standard derivation in cause of western blot and gene expression experiments?
Response 4: The results were expressed as the relative expression of protein to control group.
Point 5: Please add a quantitative analysis in bars for the immunohistochemistry experiments. Moreover, add the statistical analysis when claiming that “… KAE treatment inhibited the expression of Iba-1 in the striatum tissues of LPS-induced mice. This finding indicates that KAE inhibits the expression of Iba-1 following stimulation of mice by LPS…”, as you only show the results of one mouse. In addition, please add a size bar or magnification factor for this figure.
Response 5: The size bar for the immunohistochemistry experiments were added in the revised manuscript.
Point 6: Why did you not perform immunohistochemistry with mice that received KAE20?
Response 6: We are sorry for this defect in the design of experiment.
Point 7: Please explain your animal model briefly instead of only referring to another publication. How many animals were used for the whole experiment? Why differ the “n” between 4 and 8? How was KAE dissolved? How were LPS and KAE applied? Please add that to your methods.
Response 7: Number of animals, KAE treatment, and LPS administration were supplemented in Methods section in the revised manuscript.
Point 8: What was the rationale to isolate proteins for ELISA with saline (4.5.) and for western blot with RIPA buffer (4.6.)?
Response 8: The protein samples obtained from RIPA pyrolysis solution can be used for Western blot.
Point 9: Please connect your discussion more detailed to the investigated parameters and analyses you performed. For example:
a. Discuss the difference between the two concentrations you used. Which one is “better” and why?
b. Does KAE have some negative side effects?
c. Why did you pre-treat the animals for seven days with KAE? Is it necessary for KAE to exert its protective effects? Are there other studies with different application time points or even just with one time application?
d. Please add limitations to your study.
Response 9:
In present study, the results showed that Kaempferol treatment at the dose of 20 mg/kg and 50 mg/kg both had protective effects on striatal neurons, BBB integrity and anti-neuroinflammation of striatal tissues. Moreover, the effect of KAE 50 mg/kg group was better than that of 20 mg/kg group.
During the administration of Kaempferol in this experiment, no abnormal phenomena were observed in mice.
The experimental procedure refers to previous research.
Though the present study found that Kaempferol treatment downregulated the HMGB1 protein and TLR4 expression, there is a lack of in-depth study, such as, the translocation of HMGB1 after LPS induction and after Kaempferol treatment? In the latter research, the translocation of HMGB1 after LPS induction and Kaempferol treatment will be further explored. Moreover, the distribution and regulation mechanism of HMGB1 in different region of brain, plasma, serum or CSF will be further investigated.
Minor criticisms:
Point 10: Please announce the abbreviation “DA” (line 35) an MPTP (line 171).
Response 10: Full spellings of DA (dopamine) and MPTP (1-Methyl-4-phenyl-1,2,3,6-tetrahydropyridin) have been added in the revised manuscript.
Point 11: Line 55: “…Kaempferol (KAE) has a variety of biological activities, such as anti-oxidant, anti-inflammatory, clearing the free radicals, anti-cancer, and anti-angiogenesis, and plays an important role in cancer, inflammation and cardiovascular diseases…”. Please delete “clearing free radicals” as this is included in “anti-oxidant”. The same for “anti-cancer” and “plays an important role in cancer”.
Response 11: These sentences were corrected in the revised manuscript.
Point 12: Please check for the size of the font, as it differs several times.
Response 12: The size of the font was checked and revised carefully.
Point 13: Pease add a size bar or magnification factor for figure 1 A (a-d) and 3 A (a-d).
Response 13: The size bar was added in Fig. 1A and Fig. 3A in the revised manuscript.
Point 14: Figure 1: Please use “A, B, C”, like in the description below, instead of “A,A,B”. Adjust the letters also in line 70 and 71.
Response 14: We are very sorry for this error. Figure 1 was corrected in the revised manuscript.
Point 15: Figure 2: Please describe A and B once again in the description of figure 2.
Response 15: In the legend of Figure 2, the representatives of A and B were added in the revised manuscript.
Point 16: Figure 3: You used “B” twice in the description of this figure. Please change that.
Response 16: Figure 3 was corrected in the revised manuscript.
Point 17: Line 97: Please change “occluding” to “occludin”.
Response 17: This error was corrected in the revised manuscript.
Point 18: Please choose a more representative western blot for the quantitative results of HMGB1 (figure 5C). The bars of the control and LPS+KAE20 look nearly the same, but the representative western blots were nearly showing the opposite.
Response 18: Figure 5C was replaced in the revised manuscript.
Point 19: Line 237: “…After LPS stimulating for 6 hours…”. Please change that into “Six hour after LPS application”, as otherwise, it sounds like there was a continuous LPS applicative for six hours.
Response 19: These sentences were corrected in the revised manuscript.
Point 20: Pease add the used internal control for western blot analyses in section 4.6 and be careful that –ΔΔCt is superscripted.
Response 20: The error was corrected in the revised manuscript.
Reviewer 4 Report
Review of manuscript ijms-401867
In this manuscript the authors inject LPS into the substantia nigra of mice in order to induce inflammation similar to the pathology of Parkinson’s disease. They test the protective effects of Kaempferol (KAE) in this model. They analyze the effects of LPS with and without KAE on various cell types, and also measure the effects on inflammatory markers and the permeability of the blood brain barrier. They have recently published a study in which they conducted the same, and even more analyses, in the cortex and hippocampus. In this paper they are just investigating a different brain area, so it is not as novel as their previous study. There are other issues that need to be addressed before publication can be considered.
1. There are many grammatical mistakes and therefore careful editing by a native English speaker is needed.
2. In the Introduction (last paragraph), the authors should state what KAE is, e.g. which plants is it found in? From where is it derived?
3. The figure legend in Figure 1 is not descriptive enough, and also the images are labeled improperly. This is pretty careless.
4. The legend of Figure 2 is also not detailed enough. Also, why is the label for part A KAE50+LPS but for part B it is LPS+KAE50. Please be consistent.
5. How many animals were included in each experimental group? In fact, the n for every assay should be made clear.
6. The authors state that the LPS-induced model and drug treatment were described in their previous paper (in Int. Immunopharmacol). This paper does not appear to be open access; therefore they need to describe the procedures, at least briefly, in this manuscript as well. Again, how many animals in each experiment were used? What was the vehicle used? Did control animals get an equal volume of vehicle injected? Also, in the previous paper, 25, 50, and 100 mg/kg doses of KAE were used. Why were 20 and 50 mg/kg doses used in this study?
Author Response
Point 1: There are many grammatical mistakes and therefore careful editing by a native English speaker is needed.
Response 1: The English languages were checked and revised in the revised manuscript.
Point 2: In the Introduction (last paragraph), the authors should state what KAE is, e.g. which plants is it found in? From where is it derived?
Response 2: Kaempferol is derived from the roots of the ginger plant Kaempferol galanga L.
Point 3: The figure legend in Figure 1 is not descriptive enough, and also the images are labeled improperly. This is pretty careless.
Response 3: Figure 1 was corrected in the revised manuscript.
Point 4: The legend of Figure 2 is also not detailed enough. Also, why is the label for part A KAE50+LPS but for part B it is LPS+KAE50. Please be consistent.
Response 4: Thank you for your advice. Figure 2 was corrected in the revised manuscript.
Point 5: How many animals were included in each experimental group? In fact, the n for every assay should be made clear.
Response 5: Number of animals in each experimental group was added in Methods section.
Point 6: The authors state that the LPS-induced model and drug treatment were described in their previous paper (in Int. Immunopharmacol). This paper does not appear to be open access; therefore they need to describe the procedures, at least briefly, in this manuscript as well. Again, how many animals in each experiment were used? What was the vehicle used? Did control animals get an equal volume of vehicle injected? Also, in the previous paper, 25, 50, and 100 mg/kg doses of KAE were used. Why were 20 and 50 mg/kg doses used in this study?
Response 6: According to the reviewer's suggestion, a brief procedure of animal experiment was added to the Methods section. Referring to the previous research results, 50mg/kg has a good anti-inflammatory effect, so there is no 100mg/kg dose group in this study.
Round 2
Reviewer 1 Report
Authors are addressed most of the comments.
Author Response
Point 1: Authors are addressed most of the comments.
Response 1: Thank you for reviewing our manuscript again.
Reviewer 2 Report
Please, include this point (and reference) into the Result and Introduction sections of the manuscript: "BBB integrity by Evance Blue staining was investigated in our previous article (Int. Immunopharmacol. 2018, 56: 29-35.). Therefore, in this study, we focused on the BBB integrity in the striatal tissues. The ultrastructure of BBB was observed using electron microscopy and tight junction proteins was determined by WB"
Author Response
Point 1: Please, include this point (and reference) into the Result and Introduction sections of the manuscript: "BBB integrity by Evance Blue staining was investigated in our previous article (Int. Immunopharmacol. 2018, 56: 29-35.). Therefore, in this study, we focused on the BBB integrity in the striatal tissues. The ultrastructure of BBB was observed using electron microscopy and tight junction proteins was determined by WB"
Response 1: Thank for your suggestion. These sentences were added into the Result and Introduction sections of the manuscript.
Reviewer 3 Report
Comments to the authors:
Major criticisms
The English language could still be improved. However, one thing needs to be changed. Very often, you use phrases like the following, for example:
Line 15: “In this study, the effects and possible mechanism of KAE on striatal dopaminergic neurons induced by LPS in mice were further investigated.”
Line 85: “Effect of kaempferol on the striatum of LPS-induced mice.”
Line 115: “In LPS-induced mice, levels of occluding (Fig 3B, p < 0.01), claudin-1 (Fig 3C, p < 0.01), and CX-43 (Fig 3D, p < 0.01) in the striatum of mice were all decreased significantly compared with control group.”
Line 122: “Effect of kaempferol on the ultrastructure and tight junction proteins of BBB in the striatum of mice induced by LPS.”
Line 137: “Effect of kaempferol on pro-inflammatory factors and chemokines in the 137 striatum of mice induced by LPS.”
Line 148: “Effect of kaempferol on mRNA and protein expression of HMGB1 and TLR4 in the striatum of mice induced by LPS.”
The bold marked phrases cannot be used! What are “mice induced by LPS”/”LPS-induced mice” or ”dopaminergic neurons induced by LPS”? You have to use phrases, such as:
- LPS induced damage/injury
- LPS-treated mice
- mice damaged/injured by LPS or others.
“Point 4: Why do your control groups have no standard derivation in cause of western blot and gene expression experiments?
Response 4: The results were expressed as the relative expression of protein to control group.”
à Even through the results are shown as relative expressions, there still can exist a standard derivation. Did you compare the expression of your 3 control samples with each other? Were they quite similar? The differences between the controls is an important point. In my opinion, this should be discussed in more detail.
“Point 5: Please add a quantitative analysis in bars for the immunohistochemistry experiments. Moreover, add the statistical analysis when claiming that “… KAE treatment inhibited the expression of Iba-1 in the striatum tissues of LPS-induced mice. This finding indicates that KAE inhibits the expression of Iba-1 following stimulation of mice by LPS…”, as you only show the results of one mouse. In addition, please add a size bar or magnification factor for this figure.
Response 5: The size bar for the immunohistochemistry experiments were added in the revised manuscript.”
Why did the authors not address the bold marked points?
“Response 2: In this study, microglia activation was observed qualitatively…”. The authors answered another reviewer (Response 2) that the activation of the microglia was qualitatively observed. What was the reason for this? The authors should not leave this question unanswered. I agree with the other reviewer that the quantification should be provided to support the results with more “n” instead of one. In this case, don’t forget to add “n”.
“Point 6: Why did you not perform immunohistochemistry with mice that received KAE20?
Response 6: We are sorry for this defect in the design of experiment.”
à You should give a short explanation of this in your manuscript. Otherwise, it may confuses the reader.
Even through the discussion is changed it appears more like a narrative text instead of a discussion. The points mentioned under “Point 9” in the review were intended as some examples to gain a better connection. Therefore, they should have been included into the discussion. The discussion still needs a better and more comprehensible connection to the results.
Minor criticisms
Line 56: change “is“ to “are”.
Line 65: delete “etc”.
Line 90: “Kaempferol inhibits LPS-induced the microglia activation in striatum of mice.”
Add (a), (b), (c) to figure 3A.
“Point 16: Figure 3: You used “B” twice in the description of this figure. Please change that.
Response 16: Figure 3 was corrected in the revised manuscript.”
à The mistake was made in the figure description and it was not corrected (“A,B,B,C” instead of “A,B,C,D”).
Please describe the abbreviation “TJ”.
“Point 14: Figure 1: Please use “A, B, C”, like in the description below, instead of “A,A,B”. Adjust the letters also in line 70 and 71.
Response 14: We are very sorry for this error. Figure 1 was corrected in the revised manuscript.”
à Unfortunately, the authors did not change the letters in the results. Now line 81: “However, KAE can significantly improve the expression level of TH (Figure 1BA) and PSD95 (Figure 1CB) compared to the LPS-induced group (p < 0.05).”
Author Response
Major criticisms
Point 1: The English language could still be improved. However, one thing needs to be changed. Very often, you use phrases like the following, for example:
Line 15: “In this study, the effects and possible mechanism of KAE on striatal dopaminergic neurons induced by LPS in mice were further investigated.”
Line 85: “Effect of kaempferol on the striatum of LPS-induced mice.”
Line 115: “In LPS-induced mice, levels of occluding (Fig 3B, p < 0.01), claudin-1 (Fig 3C, p < 0.01), and CX-43 (Fig 3D, p < 0.01) in the striatum of mice were all decreased significantly compared with control group.”
Line 122: “Effect of kaempferol on the ultrastructure and tight junction proteins of BBB in the striatum of mice induced by LPS.”
Line 137: “Effect of kaempferol on pro-inflammatory factors and chemokines in the 137 striatum of mice induced by LPS.”
Line 148: “Effect of kaempferol on mRNA and protein expression of HMGB1 and TLR4 in the striatum of mice induced by LPS.”
The bold marked phrases cannot be used! What are “mice induced by LPS”/”LPS-induced mice” or ”dopaminergic neurons induced by LPS”? You have to use phrases, such as:
- LPS induced damage/injury
- LPS-treated mice
- mice damaged/injured by LPS or others.
Response 1: These languages were checked and revised carefully.
Point 2: “Point 4: Why do your control groups have no standard derivation in cause of western blot and gene expression experiments? Response 4: The results were expressed as the relative expression of protein to control group.” à Even through the results are shown as relative expressions, there still can exist a standard derivation. Did you compare the expression of your 3 control samples with each other? Were they quite similar? The difference between the controls is an important point. In my opinion, this should be discussed in more detail.
Response 2: In western blot experiments, the protein band intensities were normalized to β-actin. The results from animals under various experiment conditions then were normalized by value of the corresponding control animal (fold change relative to control). In gene expression experiments, the results from various experiment groups were normalized by value of the corresponding control (fold change relative to control).
Point 3: “Point 5: Please add a quantitative analysis in bars for the immunohistochemistry experiments. Moreover, add the statistical analysis when claiming that “… KAE treatment inhibited the expression of Iba-1 in the striatum tissues of LPS-induced mice. This finding indicates that KAE inhibits the expression of Iba-1 following stimulation of mice by LPS…”, as you only show the results of one mouse. In addition, please add a size bar or magnification factor for this figure. Response 5: The size bar for the immunohistochemistry experiments were added in the revised manuscript.” Why did the authors not address the bold marked points? “Response 2: In this study, microglia activation was observed qualitatively…”. The authors answered another reviewer (Response 2) that the activation of the microglia was qualitatively observed. What was the reason for this? The authors should not leave this question unanswered. I agree with the other reviewer that the quantification should be provided to support the results with more “n” instead of one. In this case, don’t forget to add “n”. “Point 6: Why did you not perform immunohistochemistry with mice that received KAE20? Response 6: We are sorry for this defect in the design of experiment.” à You should give a short explanation of this in your manuscript. Otherwise, it may confuse the reader.
Response 3: The quantitative analysis and a size bar were added in Figure 2.
Point 4: Even through the discussion is changed it appears more like a narrative text instead of a discussion. The points mentioned under “Point 9” in the review were intended as some examples to gain a better connection. Therefore, they should have been included into the discussion. The discussion still needs a better and more comprehensible connection to the results.
Response 4: The discussion section has been further optimized and condensed.
Point 5: Minor criticisms
Line 56: change “is” to “are”.
Line 65: delete “etc”.
Line 90: “Kaempferol inhibits LPS-induced the microglia activation in striatum of mice.”
Add (a), (b), (c) to figure 3A. “Point 16: Figure 3: You used “B” twice in the description of this figure. Please change that. Response 16: Figure 3 was corrected in the revised manuscript.” à The mistake was made in the figure description and it was not corrected (“A,B,B,C” instead of “A,B,C,D”).
Please describe the abbreviation “TJ”.
“Point 14: Figure 1: Please use “A, B, C”, like in the description below, instead of “A,A,B”. Adjust the letters also in line 70 and 71. Response 14: We are very sorry for this error. Figure 1 was corrected in the revised manuscript.” à Unfortunately, the authors did not change the letters in the results. Now line 81: “However, KAE can significantly improve the expression level of TH (Figure 1BA) and PSD95 (Figure 1CB) compared to the LPS-induced group (p < 0.05).”
Response 5: These errors were corrected in the revised manuscript.
Reviewer 4 Report
Although more detail could still be given in the results and methods sections, this revision is a major improvement on the first draft. However, there are still many grammatical mistakes for example, and I find it difficult to believe that the paper was reviewed by a native English speaker. Just a few examples are:
In the abstract: on lines 17 and 20, 'and' needs to be added at appropriate places in the sentences otherwise they are difficult to interpret.
On line 24 (in the abstract) and line 42, 'may be have' should be 'may have'
On lines 36-37, Parkinson's disease should be abbreviated, as it is defined earlier in the paragraph on line 32.
There are too many other mistakes like this for me to list.
Author Response
Point 1: Although more detail could still be given in the results and methods sections, this revision is a major improvement on the first draft. However, there are still many grammatical mistakes for example, and I find it difficult to believe that the paper was reviewed by a native English speaker. Just a few examples are:
In the abstract: on lines 17 and 20, 'and' needs to be added at appropriate places in the sentences otherwise they are difficult to interpret.
On line 24 (in the abstract) and line 42, 'may be have' should be 'may have'
On lines 36-37, Parkinson's disease should be abbreviated, as it is defined earlier in the paragraph on line 32.
There are too many other mistakes like this for me to list.
Response 1: These languages were checked and revised carefully.
Round 3
